# Complementarity of X-, C-, and L-band SAR Backscatter Observations to Retrieve Forest Stem Volume in Boreal Forest

**Maurizio Santoro [1],\*, Oliver Cartus [1], Johan E. S. Fransson [2] and Urs Wegmüller [1]**

1   Gamma Remote Sensing, Worbstrasse 225, 3073 Gümligen, Switzerland
2   Department of Forest Resource Management, Swedish University of Agricultural Sciences, SE-901 83 Umeå, Sweden
\*   Correspondence: santoro@gamma-rs.ch; Tel.: +41-31-9517005

**Abstract:** The simultaneous availability of observations from space by remote sensing platforms operating at multiple frequencies in the microwave domain suggests investigating their complementarity in thematic mapping and retrieval of biophysical parameters. In particular, there is an interest to understand whether the wealth of short wavelength Synthetic Aperture Radar (SAR) backscatter observations at X-, C-, and L-band from currently operating spaceborne missions can improve the retrieval of forest stem volume, i.e., above-ground biomass, in the boreal zone with respect to a single frequency band. To this scope, repeated observations from TerraSAR-X, Sentinel-1 and ALOS-2 PALSAR-2 from the test sites of Remningstorp and Krycklan, Sweden, have been analyzed and used to estimate stem volume with a retrieval framework based on the Water Cloud Model. Individual estimates of stem volume were then combined linearly to form single-frequency and multi-frequency estimates. The retrieval was assessed at large 0.5 ha forest inventory plots (Remningstorp) and small 0.03 ha forest inventory plots (Krycklan). The relationship between SAR backscatter and stem volume differed depending on forest structure and environmental conditions, in particular at X- and C-band. The highest retrieval accuracy was obtained at both test sites at L-band. The combination of stem volume estimates from data acquired at two or three frequencies achieved an accuracy that was superior to values obtained at a single frequency. When combining estimates from X-, C-, and L-band data, the relative RMSE for the 0.5 ha inventory plots at Remningstorp was 31.3%. For the 0.03 ha inventory plots at Krycklan, the relative RMSE was above 50%. In a retrieval scenario involving short wavelength SAR backscatter data, these results suggest combining multiple frequencies to ensure the highest possible retrieval accuracy achievable. Retrievals should be undertaken to target spatial scales well above the size of a pixel.

**Keywords:** SAR backscatter; TerraSAR-X; Sentinel-1; ALOS-2 PALSAR-2; Water Cloud Model; stem volume; above-ground biomass

## 1. Introduction

The increasing number of orbiting platforms acquiring an unprecedented number of images of the Earth's land surfaces fosters mapping and monitoring applications that cannot be achieved in timely manner with in situ observations. The way the Earth land surfaces are perceived changes as a more global perspective is being achieved, thus unraveling the systemic aspects of the bio- and geophysical processes over land. In this respect, observations of forest extent and density are of utmost importance [1] because forests control the carbon cycle, which ultimately feeds back to climate [2].

Synthetic Aperture Radar (SAR) instruments comply with the requirement of continuous observations because the data acquired remains unaffected by cloud cover and solar illumination.

Yet, the signal recorded by radar represents the interplay of geometric and dielectric properties of the objects seen on the ground. For example, the SAR backscattered intensity from a forest combines (i) structural information related to the arrangement of the trees on the ground and the trees architecture, (ii) dielectric information related to the water content of the tree, (iii) properties of the soil underneath the forest in terms of surface roughness and moisture, and (iv) additional specific conditions of the environment at the time of image acquisition (e.g., presence of snow cover). The interplay of such factors depends on the frequency band at which the radar operates, the viewing geometry and the polarization of the microwave. The development of thematic mapping applications based on SAR data, therefore, requires careful consideration of these multiple factors.

The retrieval of forest variables, in particular above-ground biomass, is a major topic of investigation with SAR because of the sensitivity of the observed signal to vegetation structure; accordingly a rather wide range of approaches have been presented in literature to improve the estimation of biomass with respect to past studies and overcome the major issue with any remote sensing observable when estimating biomass, i.e., the fact that biomass cannot be directly measured with remote sensing [3].

One line of biomass retrieval approaches aims at disentangling the multiple components affecting the SAR signal by modeling individual terms and removing the modeled components not related to biomass. This is viable when the SAR observable is dominated by one of the factors listed above. In the case of interferometric data in a repeat-pass scenario (InSAR), it is attempted to compensate for temporal decorrelation with a model in order to maximize the volumetric decorrelation, e.g., as in [4], in order to univocally relate to tree height, i.e., a major predictor of forest above-ground biomass (AGB).

In the case of observations of the radar backscattered intensity, referred to as SAR backscatter from here onwards, the information content on the parameter of interest in the measurement is significantly masked by the different factors listed above. As a result, disentangling signal contributions would introduce significant uncertainties in the compensated observation. A more viable approach in this case is to rely on an extended vector of SAR observations where a diversity of information is captured in the signals. For example, a multi-temporal dataset of observations can improve the retrieval of forest biomass with respect to a retrieval based on a single observation [5,6]. The information content on biomass in each observation is maximized by combining estimates from individual images in a way that estimates from an image dataset characterized by higher sensitivity to biomass are preferred to values estimated from image where apparently there is no sensitivity of the biomass to the backscatter [3]. With a multi-temporal dataset, systematic issues persist such as lack of sensitivity of the observable to biomass above a certain level as in the case of short wavelength SAR backscatter data [6,7]. To further improve the retrieval, one could exploit the frequency dimension, which in recent times has become possible thanks to the availability of repeated SAR observations from spaceborne platforms operating at different wavelengths.

The potential of multi-frequency observations to retrieve biomass has been reported in studies investigating mostly a small number of airborne observations acquired during the 1990s [8–12]. Studies focused on C-, L-, and P-band observations to retrieve AGB in temperate forests, concluding that the retrieval accuracy increased at lower frequency. Since the investigations were exploratory, it could not be concluded what is the actual benefit of combining observations from multiple frequencies to retrieve forest biomass. More recently, the combination of multiple C-, L-, and P-band observations from the Remningstorp test site in Sweden (see also Section 2) has been investigated to appraise the contribution of these three frequencies to the retrieval of biomass [13]. While it was confirmed that P-band has the strongest sensitivity to biomass compared to C- and L-band, the results indicated that the retrieval of AGB with multi-temporal C- and L-band data achieved comparable accuracies as a single P-band observation up to ~150 tons/ha [13].

With the simultaneous availability of repeated SAR observations from spaceborne SAR platforms at X-, C-, and L-band [13–16], as well as the recent launch of the NovaSAR platform carrying an S-band

sensor and the first P-band sensor in space [17] in the nearest future, the benefit of multi-frequency SAR observations in the context of biomass estimation need to be further understood as to appraise the relevance of multi-frequency observations of the SAR backscatter with respect to other observables derived from the measurements by these sensors.

The scope of this study is to advance knowledge on the retrieval of forest biomass with multi-frequency SAR data by exploring the combination of observations of short wavelength SAR backscatter (i.e., acquired at X-, C-, and L-band) over boreal forest and assessing the complementarity of observations in joint retrievals. Differently than in the exploratory studies cited above, where a small number of fully polarimetric images were available, here we profit from repeated observations at each individual frequency band and investigated the potential of combined multi-frequency and multi-temporal observations. The study has been undertaken at two test sites in Sweden where we expected that, at each frequency band, the SAR backscatter has sensitivity to biomass, even if not throughout the entire range of biomass, and the functional dependency of the SAR backscatter upon biomass differs because of the different penetration of the microwave into the forest canopy at X-, C-, and L-band. Structural differences of the forests at the two sites allows for a deeper understanding of multi-frequency signatures of SAR backscatter and to assess the impact on the retrieval of biomass.

Sections 2 and 3 describe the test sites and the SAR datasets, respectively. The model relating the SAR observations to biomass is presented in Section 4. Here, biomass is represented by the stem volume (unit: $m^3$/ha), i.e., the volume of tree trunks per unit area. Stem volume is the major predictor of biomass in northern forests and a relevant forest variable in terms of forest management. Section 5 presents the retrievals of stem volume for (i) single images, (ii) combinations of estimates of stem volume from multiple images acquired at a single frequency band and (iii) combinations of estimate from multiple images acquired in multiple frequency bands. The results are discussed in Section 6 and put into perspective with a comparison to retrievals obtained at the test sites using other remote sensing datasets. Conclusions centered on the usefulness of multi-frequency SAR backscatter data to retrieve biomass are reported in Section 7.

## 2. Test Sites

Retrieval of forest stem volume has been investigated in Sweden at the hemi-boreal forest site of Remningstorp and the boreal forest sites of Krycklan (Figure 1). At both sites, the forests are regularly inventoried with different techniques (forest field measurements, laser scanning). Measurements of forest variables are then used to assess novel retrieval methodologies based on remote sensing observations, e.g., [7,18–22].

The Remningstorp test site (Figure 1) is located in the south of Sweden (58°30′N, 13°40′E) within the transition zone from the boreal to the temperate biome. The topography is fairly flat with a ground elevation between 120 m and 145 m above sea level. The test site covers about 1200 ha of productive forest land. Prevailing tree species are Norway spruce (*Picea abies*), Scots pine (*Pinus sylvestris*) and, less represented, birch (*Betula* spp.). Pine forests are characterized by fewer but bigger trees and a more open canopy structure compared to spruce forests of similar overall above-ground biomass. This is a result of different management practices [13]. In addition, pine forests were characterized by sandy soils whereas spruce and birch forests grow on till. Stem volume measurements were available for 48 forest field inventory plots. Each plot had a radius of 40 m, corresponding to an area of approximately 0.5 ha, and was inventoried in 2014. At each plot, trees were callipered at breast height and tree heights were measured for a sub-sample of trees. The plot locations were measured using differential GPS with post-processing producing sub-meter accuracy. Stem volumes were between 58 and 691 $m^3$/ha, with an average of 320 $m^3$/ha, corresponding to 163 Mg/ha of above-ground biomass under the assumption of a biomass conversion and expansion factor of 0.51 [23].

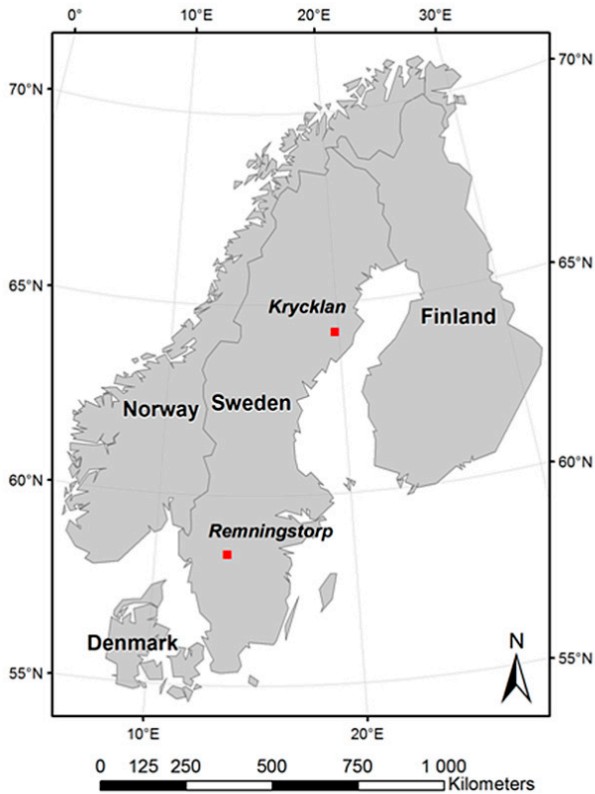

**Figure 1.** Map of Sweden showing the location of the test sites of Remningstorp and Krycklan.

The Krycklan test site is located in the north of Sweden (64°14′N, 19°50′E) and is a watershed managed and owned by both Swedish forest companies and private owners. Topography is hilly with several gorges and the ground elevation ranges between 125 m and 350 m above sea level. The size covers about 6800 ha of mainly coniferous forests. The prevailing tree species is Norway spruce, with frequent patches of Scots pine and some deciduous tree species, e.g., birch (*Betula pubescens*). The dominant soil type is till. Stem volumes were measured at 325 forest inventory plots, each with a radius of 10 m, corresponding to an area of approximately 0.03 ha. Stem volumes were between 2 and 649 m³/ha, with an average of 158 m³/ha, corresponding to 81 Mg/ha of above-ground biomass.

For each site, weather data were gathered to support the interpretation of the SAR observables and the retrieval statistics. Weather data included temperature, humidity, snow cover, precipitation and wind speed collected at several weather stations located nearby the sites (less than 10 km away). Measurements were available on a 10-, 30-, or 60-minute basis.

## 3. SAR Datasets

The datasets of SAR backscatter observations were obtained from images acquired by TerraSAR-X (X-band, wavelength of 3.1 cm), Sentinel-1A (C-band, wavelength of 5.6 cm), and Advanced Land Observing Satellite-2 (ALOS-2) Phased Array type L-band SAR-2 (PALSAR-2) (L-band, wavelength of 23.4 cm). Details on number of observations and time span of the acquisitions are provided in Tables 1 and 2 for Remningstorp and Krycklan, respectively. The time interval was selected to obtain a multi-temporal dataset of observations at each frequency band. Frequent observations by TerraSAR-X were possible thanks to dedicated observation strategy over each test site, aiming at the development on interferometric techniques to estimate height and biomass [24]. The high rate of acquisitions over Europe by Sentinel-1 explains the large number of images available. The dataset of ALOS-2 PALSAR-2 images was among the largest that could be achieved within one year (personal communication, Åke Rosenqvist). The multiple viewing geometries under which each test site was observed by TerraSAR-X and ALOS-2 PALSAR-2 resulted in a rather broad range of look angles.

**Table 1.** Synthetic Aperture Radar (SAR) dataset covering the test site of Remningstorp.

| Band | Sensor | Polarization | Look Angle | Data Sets | Time Interval |
|---|---|---|---|---|---|
| X | TerraSAR-X | Single-pol (HH or VV) | 22°–51° | 62 | 201410–201510 |
| C | Sentinel-1A | Dual-pol (VV, VH) | 39° | 33 | 201410–201510 |
| L | ALOS-2 PALSAR-2 | Dual pol (HH, HV) Full pol. (HH, HV, VV) | 28°–36° | 24 | 201409–201510 |

**Table 2.** SAR dataset covering the test site of Krycklan.

| Band | Sensor | Polarization | Look Angle | Data Sets | Time Interval |
|---|---|---|---|---|---|
| X | TerraSAR-X | Single-pol (HH or VV) | 19°–48° | 21 | 201407–201510 |
| C | Sentinel-1A | Dual-pol (VV, VH) | 39° | 78 | 201410–201510 |
| L | ALOS-2 PALSAR-2 | Dual pol (HH, HV) Full pol. (HH, HV, VV) | 28°–36° | 15 | 201408–201510 |

*SAR Data Processing*

The SAR images were ordered in Single Look Complex (SLC) format except for Sentinel-1, in which case the Ground Range Detected (GRD) format was preferred to limit processing time and given the high quality of the GRD data sets in terms of radiometric and geometric precision compared to SLC data. Each image was processed to form a terrain geocoded stack of co-registered SAR backscatter images. The processing sequence is outlined in the flowchart of Figure 2.

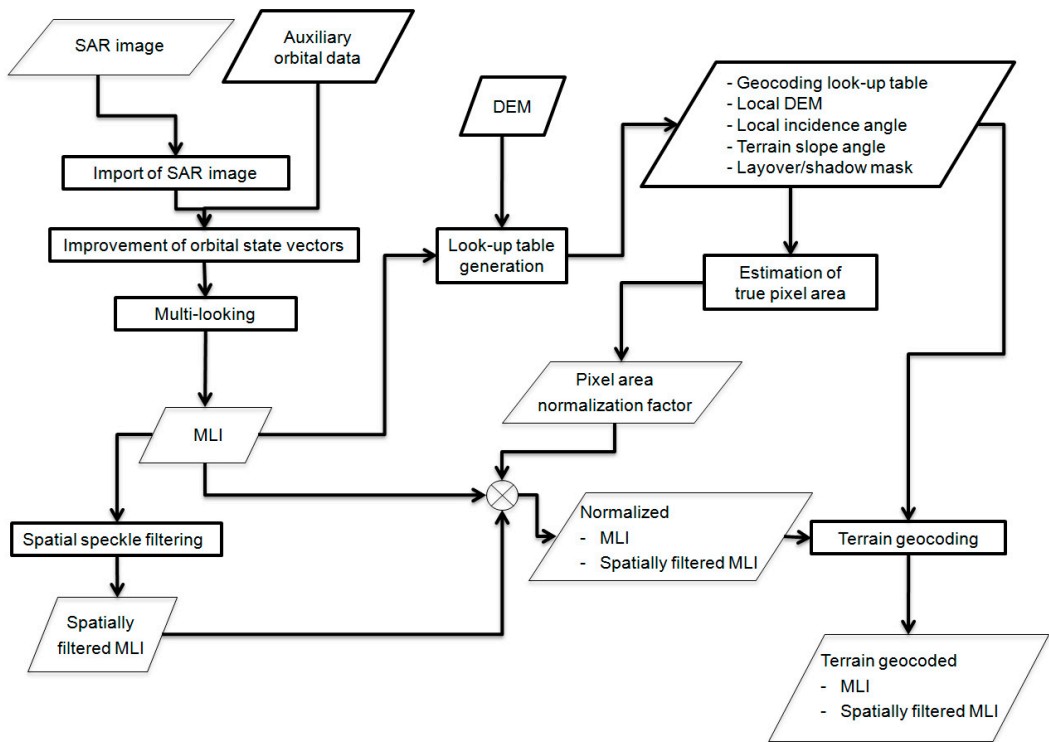

**Figure 2.** Flowchart of processing for a SAR image. DEM: Digital Elevation Model. MLI: Multi-looked Intensity.

Each SAR image was first imported in the processing environment [25]. Images available as SLC were detected to form image of the SAR backscattered intensity. Thereafter, calibration and noise reduction were applied with the calibration gain and the noise factors reported in the original

image metadata and auxiliary data files for TerraSAR-X and Sentinel-1 data. Precise orbit information was used to replace state vectors provided in the original metadata of the image in the case of Sentinel-1 images.

Taking into account that the lowest resolution was the range resolution of the ALOS-2 PALSAR-2 data, it was decided to set up the processing to obtain geocoded images with a pixel size of 20 m × 20 m. Hence, each image was first multi-looked to approximately the same pixel size, which also decreased speckle noise. This image is referred to as multi-looked intensity (MLI). We have quantified the speckle noise with an estimate of the Equivalent Number of Looks (ENL) [26].

$$ENL = \mu^2/\sigma^2, \tag{1}$$

where $\mu^2$ and $\sigma^2$ represent the squared mean value of the backscattered intensity of a target and its variance. The ENL is commonly estimated by computing the mean and the variance of the backscatter within a polygon including an area of homogeneous scattering (e.g., a field or a forest stand). In Table 3, the ENL is reported in the form of averages of estimates from several polygons characterized by homogeneous backscatter for the SAR backscatter images obtained after multi-looking and after an additional multi-channel filtering, which is described further on in this Section.

**Table 3.** Estimates of Equivalent Number of Looks (ENL) for each of the spaceborne SAR sensors used in this study.

| Sensor | ENL After Multi-Looking | ENL After Multi-Channel Filtering |
|---|---|---|
| TerraSAR-X | 12 | 168 |
| Sentinel-1 | 5 | 40 |
| ALOS-2 PALSAR-2 | 10 | 20 |

The transformation of a SAR image from radar to map geometry was implemented in the form of a geocoding look-up table [27]. For each SAR image, the look-up table was created with the aid of orbital parameters and SAR image processing parameters and elevation information in a Digital Elevation Model (DEM). In this study, we used the freely available Swedish national DEM grid 50+ (https://www.lantmateriet.se/en/Maps-and-geographic-information/Elevation-data-/GSD-Hojddata-grid-50-/), oversampled to 20 m. Because of the gentle topography at both sites, this oversampling was assumed to have negligible impact on the processing. To compensate for the geo-location error in the look-up table due to imprecise image parameters or orbital parameters, offsets were estimated by means of a cross-correlation technique between the SAR image and a simulated SAR image obtained from the DEM. Offsets were estimated at multiple positions throughout the area covered by the SAR image in order to capture possible dependencies of the offsets upon range and azimuth position. The offsets estimates were then used in a least squares regression to estimate the parameters of a polynomial describing the shift to be applied to each pixel of the look-up table in order to then match the output geometry:

$$y = a + b * range\_offset + c * azimuth\_offset. \tag{2}$$

The co-registration error described by the standard deviation of the residual shift was mostly below 0.4 times the pixel size, i.e., <10 m. With the refined look-up table, the image in the SAR geometry was projected onto the output geometry.

To compensate for slope-induced modulation of the SAR backscatter, we computed a normalization factor that accounted for the true size of the pixel instead of the size of the pixel on a flat terrain as assumed when generating the SLC and GRD image data products [28]. The area of each pixel in an image was estimated using the DEM, the orbital parameters in the SAR image metadata and the geocoding look-up table. Each normalized SAR image was re-projected to a pre-defined output

geometry meaning that all images formed a stack of co-registered SAR backscatter images. Accordingly, all normalized and spatially filtered images were geocoded to the same geometry.

Given the availability of multi-temporal data at each frequency band, we implemented the multi-channel filtering approach originally proposed in [29]. We used a spatially adaptive version of the filter to allow better estimates of the radar cross-section over textured terrain. For this reason, the multi-channel filter was driven with a spatially filtered version of the multi-looked SAR image obtained with the texture-based GAMMA MAP filter [30]. For the SAR backscatter images obtained after the multi-channel filtering, the ENL increased substantially (Table 3). The largest ENL of 168, corresponding to a residual noise of 0.32 dB was obtained for TerraSAR-X data thanks to the substantially higher spatial resolution compared to Sentinel-1 and ALOS-2. The ENL of 40 for the Sentinel-1 dataset, corresponding to 0.64 dB, was a result of the very dense stack of observations (Table 1). On the contrary, the ENL of 20 for ALOS-2 PALSAR-2 images, corresponding to 0.88 dB, was a consequence of the rather small number of images available and the strong correlation of observations in time [31].

To each forest inventory plot, we associated the area-weighted mean value of the SAR backscatter of pixels located within the perimeter. For the 0.5 ha inventory plots of Remningstorp, this corresponded to averaging over the area of approximately 12 pixels. For the 0.03 ha inventory plots of Krycklan, the average backscatter was computed over the area of slightly less than one pixel. Such averages formed the dataset of SAR backscatter observations used to train the retrieval models and estimate stem volume.

## 4. Methods

The SAR backscatter was expressed as a function of the stem volume of a forest with a Water Cloud Model (Equation 3). The Water Cloud Model (WCM) is a rather slim formulation of the scattering physics in a vegetated layer expressed in the form of a small number of model parameters. The drawback is that a forest is highly idealized, thus the vegetation structure is not entirely represented by the model. A physically based parametric model was preferred to an empirical regression and non-parametric models because two of the objectives of this study were to understand the relationship between remote sensing observations and in situ measurements, and how it varies in time and across different forest landscapes.

We used the WCM rewritten to express the total backscatter as a function of stem volume, $V$, [32]:

$$\sigma^0_{for} = \sigma^0_{gr}e^{-\beta V} + \sigma^0_{veg}\left(1 - e^{-\beta V}\right). \tag{3}$$

The derivation of Equation (3) is here omitted since it has been extensively discussed [32,33]. It is assumed that multiple scattering terms are negligible for the frequencies considered in this study. The coefficients $\sigma^0_{gr}$ and $\sigma^0_{veg}$ represent the backscattering coefficients of the ground and vegetation layer, respectively. The exponential in Equation (3) represents the two-way forest transmissivity. The exponent was modeled as a linear function of stem volume [34], in which the semi-empirical coefficient $\beta$ was assumed to combine gap and vegetation transmissivity properties [33].

For a given measurement of the SAR backscatter, stem volume was estimated with Equation (4), representing the inverse of the WCM in Equation (3). For the inversion, the model parameters $\sigma^0_{gr}$, $\sigma^0_{veg}$ and $\beta$ need to be estimated first.

$$V = -\frac{1}{\beta}ln\left(\frac{\sigma^0_{veg} - \sigma^0_{for}}{\sigma^0_{veg} - \sigma^0_{gr}}\right). \tag{4}$$

Particular care was taken in case a backscatter measurement is not within the range of $\sigma^0_{gr}$ and $\sigma^0_{veg}$ [33]. For backscatter observations, $\sigma^0_{for}$, smaller than the estimate of $\sigma^0_{gr}$ in case the model predicts an increase of backscatter with increasing stem volume or larger than $\sigma^0_{gr}$ in case the model predicts a decrease of backscatter, a stem volume of 0 m³/ha was assumed. For backscatter observations, $\sigma^0_{for}$,

larger than the estimate of $\sigma^0{}_{veg}$ in case the model predicts an increase of backscatter with increasing stem volume or smaller than $\sigma^0{}_{veg}$ in case the model predicts a decrease of backscatter, a stem volume equal to the maximum stem volume was assumed.

The availability of multiple observations of the SAR backscatter allowed for multiple estimates of stem volume for a given sampling unit. The accuracy of the stem volume estimates then depends on several factors such as the sensitivity of the backscatter to stem volume, the environmental conditions at image acquisition, polarization, look angle and forest structural parameters. To improve the retrieval accuracy, it is therefore advisable to combine the estimates from individual backscatter observations according to some predefined rule that maximize the contribution of estimates from images with strongest sensitivity of the backscatter to stem volume while neglecting those estimates that are virtually characterized by random noise. Here, we extended the concept of the multi-temporal combination presented in [6,7,33] to all datasets in the data pool of SAR observations

$$V_{mt} = \frac{\sum_{i=1}^{N} w_i V_i}{\sum_{i=1}^{N} w_i}. \tag{5}$$

The weights $w_i$ were set equal to the inverse of the retrieval root mean square error (RMSE) of the dataset used for training the model. The weights were furthermore reinforced with the fraction of samples having a backscatter within the range of modeled backscatter values, $p_{train}$ and $p_{test}$ [33]. This was considered to be a compact way of expressing the reliability of a dataset to retrieve stem volume

$$w_i = \frac{p_{train,i} \cdot p_{test,i}}{\left(RMSE_{train,i}\right)^2}. \tag{6}$$

To quantify the agreement between stem volumes from the dataset of in situ measurements acting as reference and values retrieved from the SAR data, we used the relative RMSE in Equation (7), i.e. the retrieval root mean square error divided by the average stem volume from the dataset of the reference measurements, and the bias in Equation (8), i.e., the difference of the averages from the sets of estimated and reference stem volumes. $M$ refers to the number of samples included in the dataset used to compute the statistics

$$relative\ RMSE = \frac{\sqrt{\frac{\sum_{i=1}^{M} \left(\hat{V}_{mt,i} - V_{ref,i}\right)^2}{M}}}{\frac{\sum_{i=1}^{M} V_{ref,i}}{M}}, \tag{7}$$

$$bias = \frac{\sum_{i=1}^{M} \hat{V}_{mt,i}}{M} - \frac{\sum_{i=1}^{M} V_{ref,i}}{M}. \tag{8}$$

## 5. Results

The complementarity of multiple short-wavelength datasets of the SAR backscatter in the context of forest biomass estimation was approached by performing a signature analysis of the backscatter at a given frequency and polarization, and through time (Section 5.1). We then investigated the properties of the WCM for each dataset (Section 5.2) and quantified the retrieval error per dataset based on a combination of estimates of stem volume (Section 5.3). For this, all samples available at each test site were used, i.e., the 48 0.5 ha inventory plots in Remningstorp and the 325 0.03 ha inventory plots in Krycklan. A separate assessment of the short-wavelength multi-frequency retrieval approach is presented and discussed in Section 5.4, where the model is trained and tested with datasets independent from each other.

*5.1. Signatures of Forest Backscatter as a Function of Stem Volume*

At first, an analysis of the relationship between the SAR backscatter and stem volume was undertaken for each frequency band and polarization available. The scope of this analysis was twofold: identify patterns in multi-frequency data and cross-check patterns at the two test sites. Although it is acknowledged that the relationship between SAR backscatter and stem volume is non-linear, an analysis of the correlation coefficient (Pearson's) gave indication on the strength and the sign of such relationship. In addition, the extensive multi-temporal and multi-frequency datasets allowed to identify broad patterns in seasonal variability and polarization configuration.

In Figure 3, we illustrate the correlation coefficient for each image as a function of day-of-year (DOY). The correlations have been grouped in terms of frequency band and polarization and presented for each test site. Additionally, we visualize dates characterized by frozen and unfrozen environmental conditions with blue and red symbols, respectively. Images acquired on days when the minimum temperature was well above the freezing point, here set to 3 °C, were labeled as unfrozen. Images acquired on days when temperature was close to the freezing point were allocated to the category of frozen conditions. Using a higher threshold, did not have any impact on the conclusions drawn for unfrozen conditions but would have created some ambiguity when interpreting the correlation coefficients of images acquired when the minimum temperature was below such higher threshold temperature.

In Remningstorp, the sensitivity of the SAR backscatter to stem volume was highest at X-band, particularly at HH- and VV-polarization (Figure 3a). The correlation coefficient was mostly below −0.5 (average: −0.67) indicating a strong decreasing trend of the SAR backscatter as a function of stem volume. The same trend was observed at cross-polarization even though it was not as marked as at HH- and VV-polarization. Frozen conditions often seemed to be characterized by a somewhat higher correlation (in absolute sense) between SAR backscatter and stem volume than unfrozen conditions, in particular for cross-polarized data, as shown by the blue and red crosses in the X-band panels of Figure 3a. At C-band, the correlation was also negative, i.e., decreasing SAR backscatter with increasing stem volume, with values between −0.5 and −0.1. Images acquired under frozen conditions were more sensitive to stem volume than data acquired under unfrozen conditions, as shown by the blue and red crosses, respectively, in Figure 3a for the C-band panels. These signatures appeared to be independent from polarization. At L-band, the correlation coefficient was high and positive, this corresponding to a marked increase of backscatter with stem volume. Somewhat higher correlation corresponding to a stronger sensitivity of the SAR backscatter to stem volume was observed for several acquisitions under unfrozen conditions compared to frozen conditions (see L-band panels in Figure 3a).

In Krycklan, the sensitivity of the X-band SAR backscatter to stem volume was weak, particularly at HH-polarization. In addition, there was no sign of a dependency of such relationship upon seasonal conditions (see X-band panels in Figure 3b). At C-band, the correlation was mostly between 0.3 and 0.5, indicating slight increase of backscatter with stem volume. This result appeared to be independent from polarization (see C-band panels Figure 3b). At L-band, the correlation coefficient was highest and positive, corresponding to a marked increase of backscatter with stem volume (see L-band panels in Figure 3b). Seasonal conditions seemed to have an effect on the relationship between L-band backscatter and stem volume, with slightly higher correlation coefficients corresponding to stronger sensitivity of the backscatter to stem volume for some of the images acquired under unfrozen conditions (red crosses).

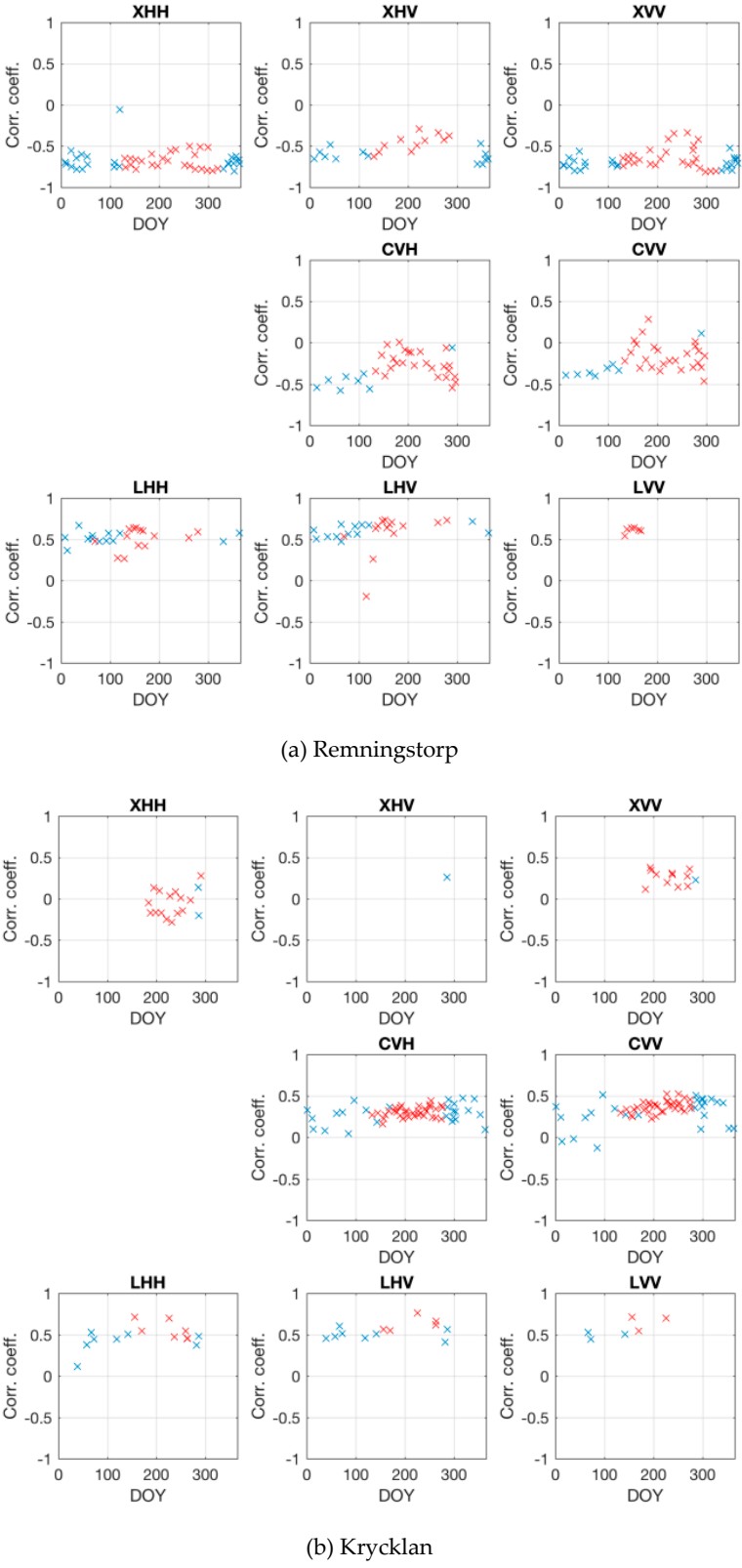

(a) Remningstorp

(b) Krycklan

**Figure 3.** Correlation coefficient of SAR backscatter and stem volume as a function of day-of-year, (DOY) at Remningstorp for 48 inventory plots (**a**) and Krycklan for 325 inventory plots (**b**). Colors refer to the minimum temperature on the day of image acquisition. Crosses are blue or red depending whether the temperature was below or above 3 °C.

## 5.2. Estimates of the WCM Parameters

Model training was undertaken by estimating the parameters $\sigma^0_{gr}$ and $\sigma^0_{veg}$ with non-linear least squares regression. For simplicity and robustness, the $\beta$ coefficient was set a priori given the large spread of the SAR backscatter measurements as a function of stem volume (Figure 4). For C- and L-band, values published in previous studies for the two sites were used, i.e., 0.0055 and 0.0042 ha/m$^3$, respectively [7,27]. For X-band, we assumed the same value as for C-band because of similar wavelengths. We did not make any distinction between frozen and unfrozen conditions since we did not have exact information on such conditions for each date of image acquisition. Reducing the number of degrees of freedom of the WCM from three to two was a sensible approach since the WCM was fitted to data characterized by a typically weak correlation.

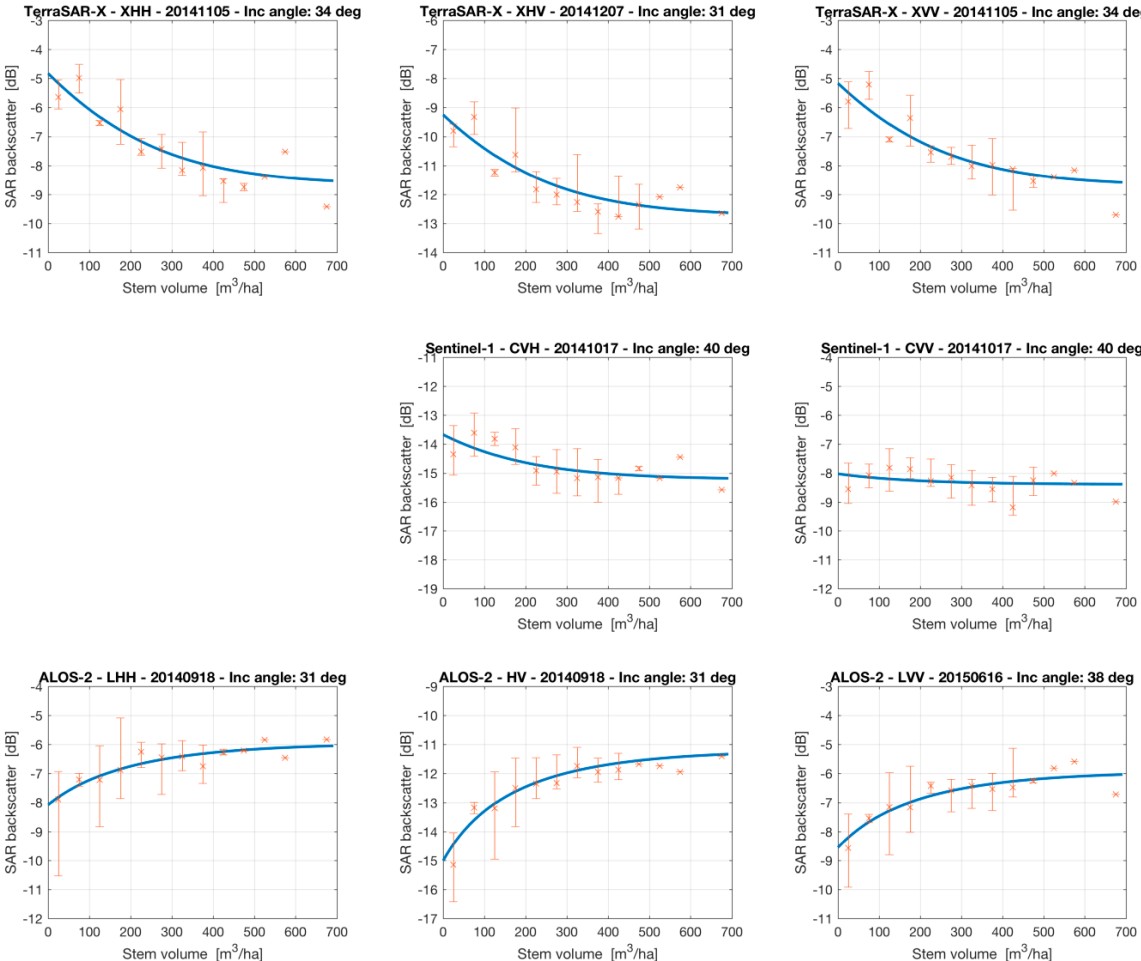

**Figure 4.** Measured and modeled SAR backscatter as a function of stem volume for the 48 0.5-ha forest plots at Remningstorp. Crosses and vertical bars represent the average and the range of SAR backscatter values in 50 m$^3$/ha wide intervals of stem volume. The blue curves represent the Water Cloud Model (WCM) fitted to the observations.

Figure 4 shows panels with one example of fitted WCM that is representative for each dataset in terms of sensor and polarization for the Remningstorp dataset. Similarly, in Figure 5, we show examples of the fitted WCM for the datasets acquired over Krycklan. The examples correspond to data acquired under unfrozen conditions (whenever possible) and for similar seasonal conditions (autumn). In Remningstorp, the relationship between SAR backscatter and stem volume changed from decreasing to increasing from X- to L-band (Figure 4). In Krycklan, instead, the backscatter always increased with increasing stem volume (Figure 5). We interpret this result as different soil properties at

the two sites. Remningstorp is characterized by predominantly peaty soils, these being wetter than soil on which pines grow, as at Krycklan so that the contribution to the total backscatter from the forest floor is enhanced. Nonetheless, the decreasing trend of the backscatter with increasing stem volume occurred under frozen and unfrozen conditions, so that it is believed that the soil surface roughness may have an important role in explaining the decreasing trend at Remningstorp as well.

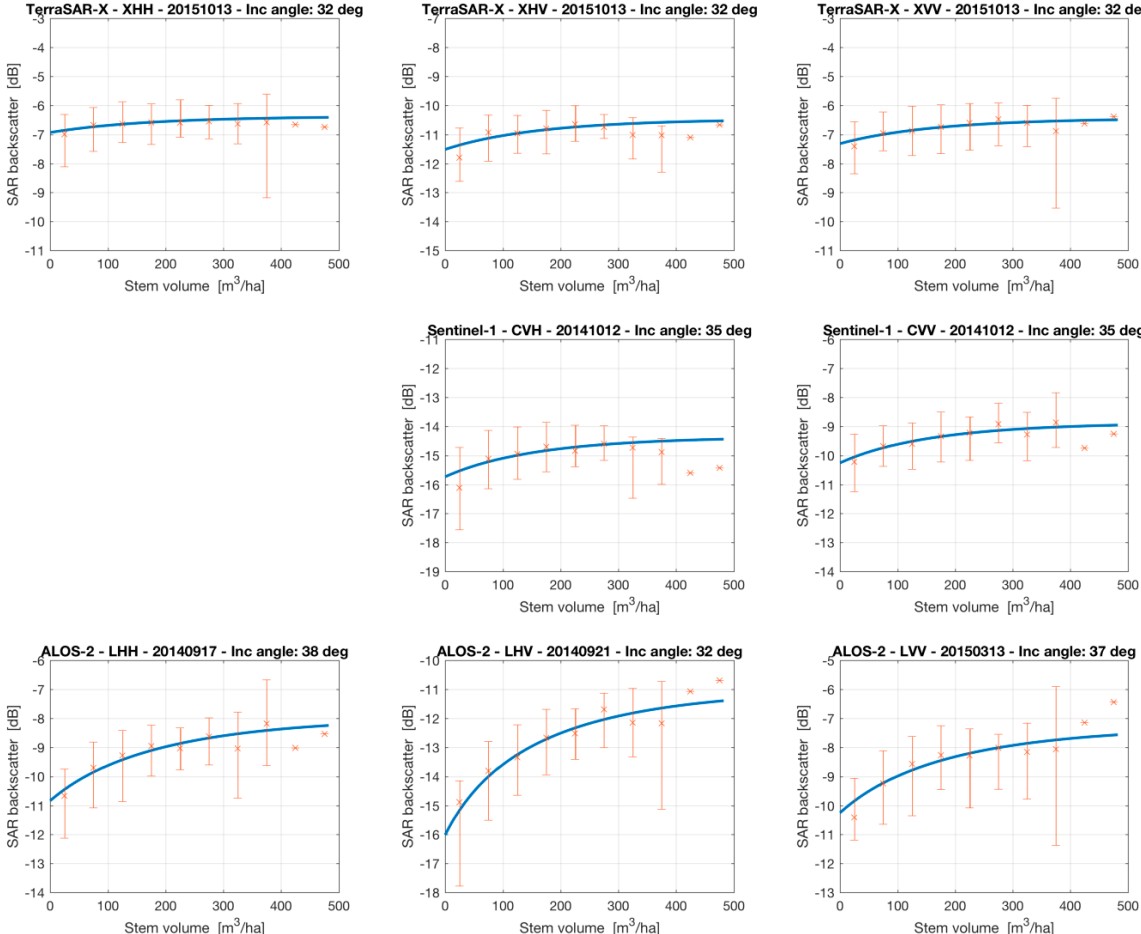

**Figure 5.** Measured and modeled SAR backscatter as a function of stem volume for the 325 0.03-ha forest plots at Krycklan. For an explanation of symbols and curves, it is referred to the caption of Figure 4.

At Remningstorp, X-band backscatter was more sensitive to stem volume than L-band backscatter (Figure 4). C-band data instead showed the weakest sensitivity to stem volume (Figure 4). More specifically, at X-band, the SAR backscatter decreased by 3–4 dB for increasing backscatter and the sensitivity of the backscatter to stem volume was very similar at all polarizations. The decreasing trend characterized the C-band data as well, even though the sensitivity of the backscatter to stem volume was smaller than at X-band; the backscatter decreased by less than 1 dB for co-polarization and by 1–2 dB for cross-polarization. At L-band, the backscatter increased with stem volume regardless of polarization; cross-polarized data showed the highest sensitivity with 3–4 dB. Taking into account that the uncertainty with the ENL values reported in Table 3 was always below 1 dB and it further reduced because of averaging observations within each of the 0.5 ha inventory plot, all trends reported here can be considered to be significant.

At Krycklan, the sensitivity of the SAR backscatter to stem volume increased with increasing wavelength (Figure 5). X- and C-band presented the weakest sensitivity to stem volume, with an increase of less than 1 dB (Figure 5). At L-band the backscatter increased steadily with increasing

stem volume by 3–4 dB for cross-polarized data and 2–3 dB for co-polarized data (Figure 5). Again, the trends appeared to be significant when relating to the ENL estimated at each frequency.

It is remarked that the spread of the observations along the model was large at all frequencies; in addition, by comparing Figure 4 (based on 0.5 ha larger plots) and Figure 5 (based on 0.03 ha large plots), one can also appreciate that the spread was smaller when the reference dataset consisted of stands or large plots. This is a valuable indication on how scales affect the retrieval or, from another point of view, at which scale the retrieval of biomass from SAR data can be considered reliable.

In an attempt to identify systematic effects explaining the spread of the backscatter observations, local incidence angle was plotted against the SAR backscatter observations, always restricting to a narrow range of stem volumes (i.e., 50 $m^3$/ha). This study was undertaken at Krycklan only because of the inventory plots in Remningstorp were located on flat terrain. The correlation coefficients between local incidence angle and SAR backscatter varied in time and were often below 0.3, thus not allowing any conclusion whether local slope could explain the variability of backscatter in this study. It needs, however, to be considered that the setting for undertaking such an experiment was sub-optimal. The SAR backscatter was taken at pixel level (20 m) while the DEM used throughout SAR processing had a spatial resolution of 50 m.

The overall behavior of the relationship between SAR backscatter and stem volume in time is summarized in Tables 4 and 5 in the form of trend indicator, dynamic range and relative root mean square error (RMSE). These parameters were derived after fitting Equation (3) for each frequency band and polarization using all samples in the reference dataset as training set. The trend indicates whether the modeled backscatter increased or decreased for increasing stem volume, i.e., $\sigma^0_{gr} < \sigma^0_{veg}$ or $\sigma^0_{gr} > \sigma^0_{veg}$. The dynamic range represents the difference $\sigma^0_{veg} - \sigma^0_{gr}$ (in absolute terms) thus being a measure of the sensitivity of the backscatter to stem volume. The relative RMSE represents the retrieval error for estimates of stem volume using Equation (7). It is remarked that the relative RMSE in Table 4 was computed using the same samples used for model training, thus not being a measure for the actual retrieval error of the methods here presented. For each frequency band and polarization, we illustrate the individual estimates of $\sigma^0_{gr}$ and $\sigma^0_{veg}$ together with measurements of temperature, precipitation and snow depth, as well as a measure of the stem volume retrieval error for each SAR backscatter image in the supplement.

**Table 4.** Multi-temporal characteristics of the relationship between SAR backscatter and stem volume at Remningstorp for a given frequency band and polarization.

| Sensor | Band-Polarization | Trend Backscatter vs. Stem Volume | Dynamic Range | Rel. RMSE |
|---|---|---|---|---|
| TerraSAR-X | XHH | Decreasing | 3–4 dB | ~50% |
| TerraSAR-X | XHV | Decreasing | 1–4 dB (*) | ≥50% |
| TerraSAR-X | XVV | Decreasing | 3–4 dB | ~50% |
| Sentinel-1 | CVH | Decreasing | 0–2 dB (*) | ~70–100% |
| Sentinel-1 | CVV | Decreasing | 0–1 dB (*) | ~90–110% |
| ALOS-2 | LHH | Increasing | 1–2 dB (**) | ~50–60% |
| ALOS-2 | LVV | Increasing | 2–3 dB (**) | ~50–60% |

(*) larger dynamic range under frozen conditions (i.e., min temperature below 0 °C)
(**) larger dynamic range under unfrozen conditions (i.e., min temperature above 0 °C)

**Table 5.** Multi-temporal characteristics of the relationship between SAR backscatter and stem volume at Krycklan for a given frequency band and polarization.

| Sensor | Band-Polarization | Trend backscatter vs. Stem Volume | Dynamic Range | Rel. RMSE |
|---|---|---|---|---|
| TerraSAR-X | XHH | Decreasing/constant | <1 dB (*) | ~110% |
| TerraSAR-X | XVV | Increasing | 1 dB | ~95% |
| Sentinel-1 | CVH | Increasing | ~1 dB | 80–100% |
| Sentinel-1 | CVV | Increasing | ~1 dB | 80–100% |
| ALOS-2 | CHH | Increasing | 3 dB | ~60% |
| ALOS-2 | CHV | Increasing | 5 dB | 50–70% |

(*) larger dynamic range under steep look angles ($< 20°$)
(**) larger dynamics range under unfrozen conditions (i.e., min temperature above 0°C)

In Remningstorp, the X-band $\sigma^0_{gr}$ was always larger than $\sigma^0_{veg}$ (Table 4 and Figure S1). The difference did not seem to be affected by seasonal or environmental conditions. The fluctuations of $\sigma^0_{gr}$ and $\sigma^0_{veg}$ were due to different look angles (Table 1) and flight directions. For the HV-polarized backscatter (Table 4 and Figure S2), the difference between $\sigma^0_{gr}$ and $\sigma^0_{veg}$ was smaller when the minimum temperature was above the freezing point, i.e., after the spring thaw at the end of April. At C-band, $\sigma^0_{gr}$ was always larger than $\sigma^0_{veg}$ under frozen conditions (i.e., when the minimum daily temperature was below or equal to 0 °C) (Table 4 and Figures S3 and S4). Under unfrozen conditions, the two model parameters were rather similar (Figures S3 and S4). As in the case of X-band, the smaller difference was related to an increase of $\sigma^0_{veg}$. At L-band, $\sigma^0_{gr}$ was always smaller than $\sigma^0_{veg}$ (Table 4 and Figures S5–S7). The very few observations under frozen conditions indicated weaker sensitivity of the backscatter to stem volume. The look angles between 28° and 36° (Table 1) did not seem to have any effect on the relationship between SAR backscatter and stem volume.

In Krycklan, images were acquired under unfrozen conditions when $\sigma^0_{gr}$ was always comparable to or smaller than $\sigma^0_{veg}$ at X-band (Table 5 and Figure S8). The dynamic range was somewhat larger at VV-polarization than HH-polarization (Figure S8). At C-band, $\sigma^0_{veg}$ was larger than $\sigma^0_{gr}$ both at VV-and VH-polarization (Table 5 and Figures S9 and S10) and the sensitivity of the SAR backscatter to stem volume remained unchanged under frozen and unfrozen conditions (Table 5). The estimates of $\sigma^0_{gr}$ were always smaller than $\sigma^0_{veg}$ at L-band, with slightly less sensitivity of the backscatter to stem volume under frozen conditions (Table 5 and Figures S11 and S12). The impact of look angle was similar to what observed at Remningstorp (Table 5).

When comparing the model parameters estimates with measurements of daily precipitation, we could not identify any systematic pattern; lack of soil moisture measurements and vegetation water content did not allow to further investigate the impact of moisture/wetness conditions on the forest backscatter. The presence of snow on the ground did not influence the backscatter except in the case of wet snow conditions, occurring under mostly unfrozen conditions, in which case the dynamic range of the backscatter was close to 0 dB.

The retrieval error was closely related to the dynamic range. Combinations of frequency band and polarization presenting the strongest sensitivity of the SAR backscatter to stem volume were also characterized by the smallest retrieval error (see Tables 4 and 5, and Figures S1–S12). The relative RMSE as a function of dynamic range are illustrated in Figure 6 for each test sites and grouped in terms of frequency band and polarization. The relationship between dynamic range and relative RMSE was practically linear, regardless of frequency band, polarization, look angle, and environmental conditions.

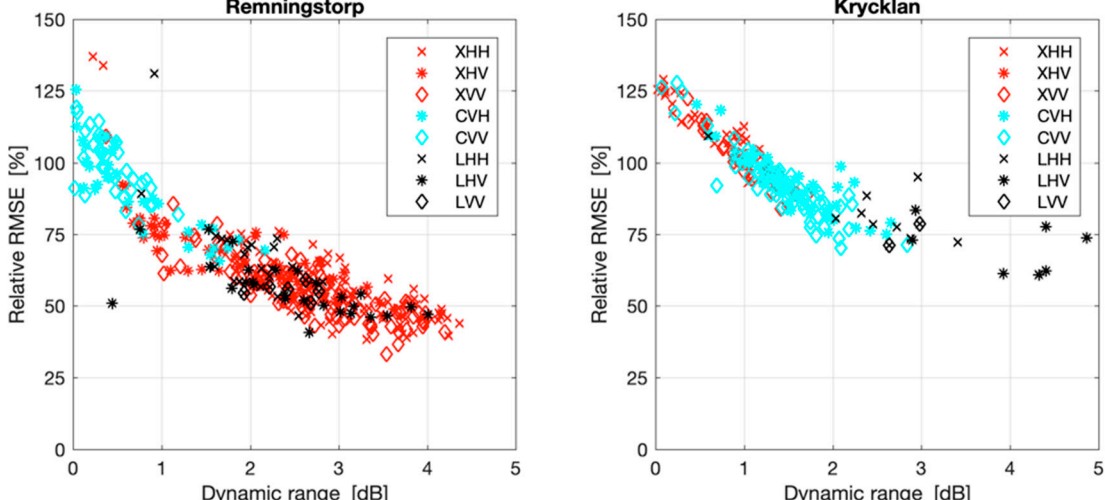

**Figure 6.** Relative root mean square error (RMSE) as a function of dynamic range at Remningstorp (left) and Krycklan (right).

### 5.3. Retrieval of Stem Volume Using Single and Multiple Observations

Once stem volume had been estimated for each SAR backscatter image, different combinations of estimates using Equation (5) were tested in order to understand the benefit of the multiple dimensions available to this study (time, polarization and frequency band). Specifically, we investigated the following combinations:

- Single frequency band, single polarization, multi-temporal data (MT combination),
- Single frequency band, multi-polarized and multi-temporal data (MTP combination),
- Multiple frequency bands, multi-polarization and multi-temporal data (MTPF combination).

Figures 7–10 show in situ and estimated stem volumes for each of these combinations. On top of each panel, the relative RMSE and the estimation bias are reported. As for the results illustrated in Tables 4 and 5 and Figures S1–S12, the test set coincided with the training set, i.e., the WCM was trained with all samples and inverted using the same samples. Since the aim of this analysis was to identify strengths and weaknesses of different types of combinations, we preferred maximizing the number of samples used to train and test the method bearing in mind that the retrieval statistics should not be interpreted as an absolute measure of the retrieval accuracy. This aspect is dealt with in Section 5.4 when the method here proposed is trained and tested with two independent datasets.

Figures 7 and 8 show the scatter plots of in situ and retrieved stem volume at Remningstorp in the case of MT retrievals for a given frequency band and polarization and MTP retrievals for a given frequency band, respectively. For CHH, the number of observations was insufficient to create a multi-temporal combination.

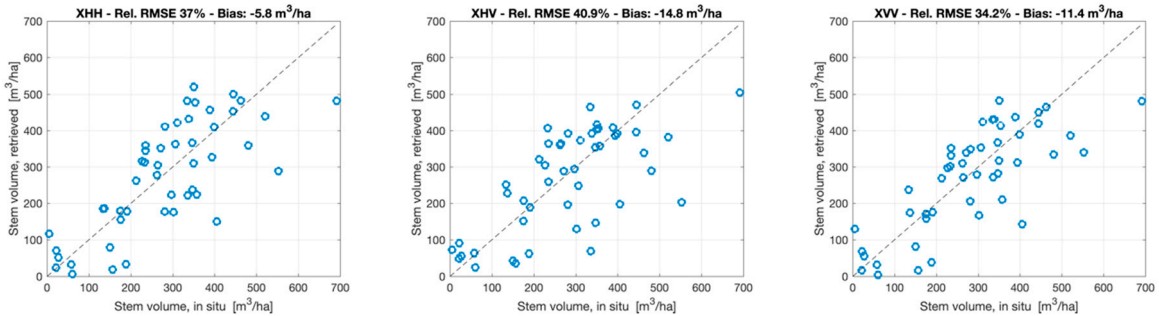

**Figure 7.** *Cont.*

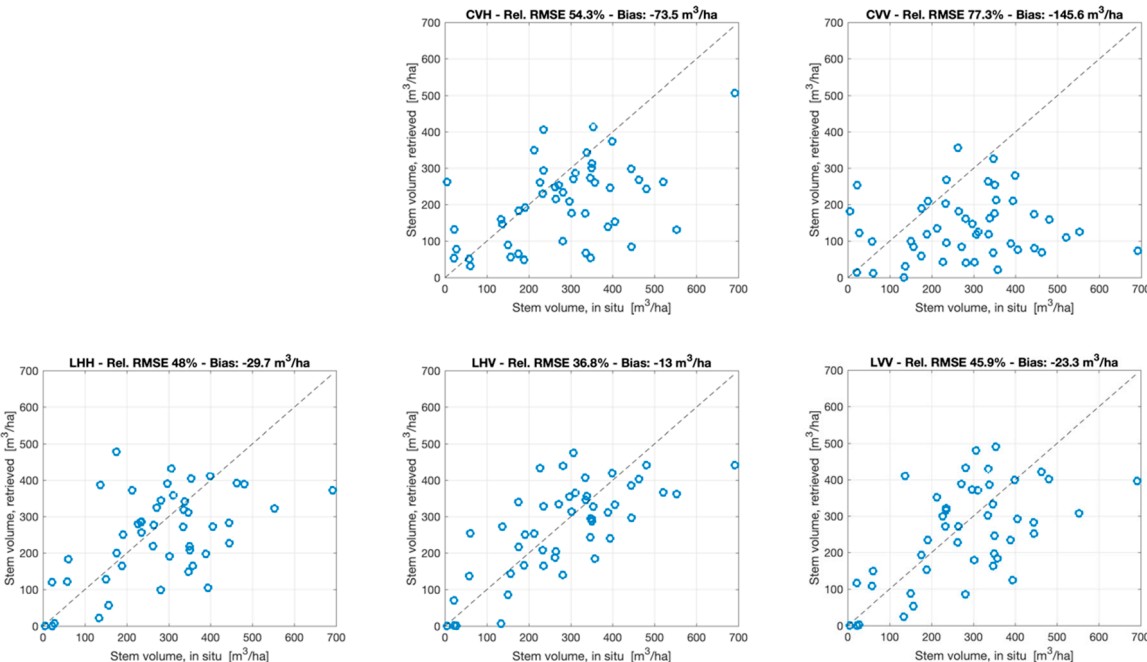

**Figure 7.** Scatter plots of measured and retrieved stem volume at Remningstorp for a multi-temporal combination of estimates obtained at a given frequency band and polarization.

Figures 9 and 10 show the scatter plots for MTPF combinations in the case of two and three frequencies, respectively. The scatter plot in Figure 10 is detailed in terms of tree species to highlight that the retrieval accuracy was influenced by tree architectures, management practices of pine, spruce and birch forests as well as the soil type (see Section 2).

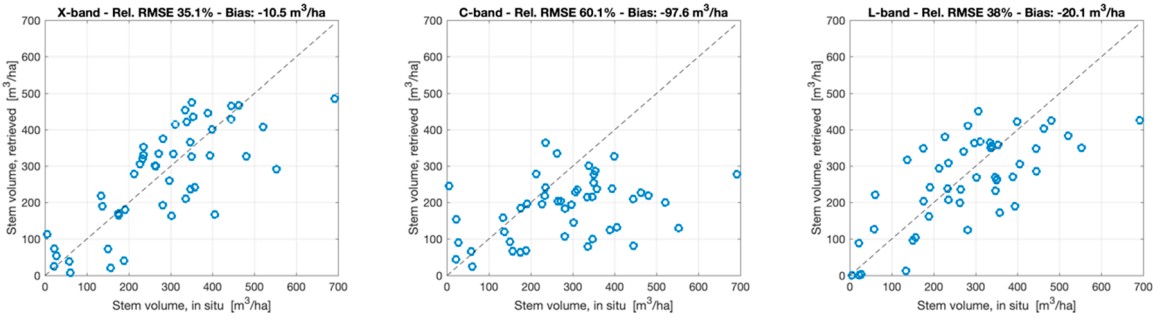

**Figure 8.** Scatter plots of measured and retrieved stem volume at Remningstorp for a multi-temporal and multi-polarization combination of estimates obtained at a given frequency band.

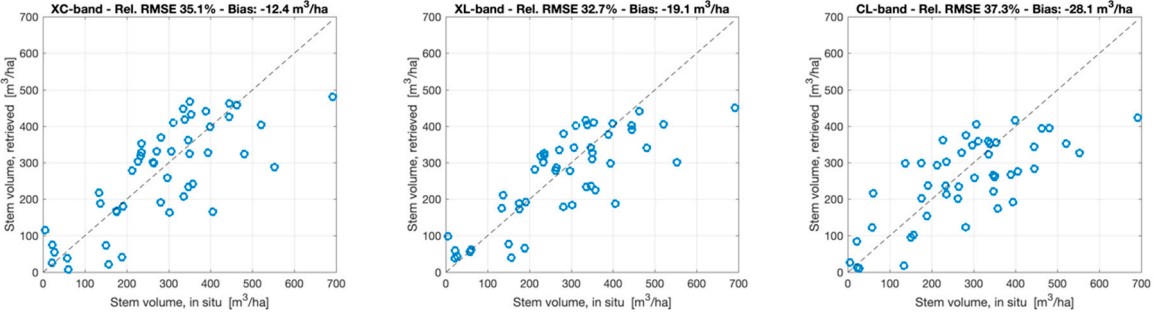

**Figure 9.** Scatter plots of measured and retrieved stem volume at Remningstorp for a combination of estimates obtained from all images acquired at two frequency bands.

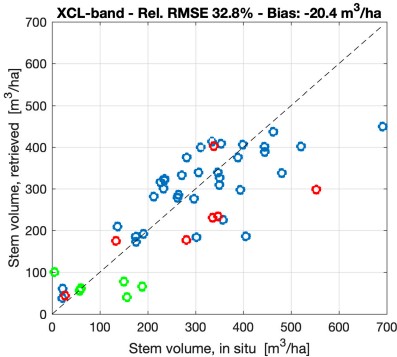

**Figure 10.** Scatter plot of measured and retrieved stem volume at Remningstorp for a combination of all estimates obtained at X-, C-, and L-band. Blue, red, and green circles refer to inventory plots with at least 50% spruce, pine, or birch trees, respectively.

At Krycklan, the agreement between in situ and retrieved stem volumes obtained with a multi-temporal combination of estimates for a single frequency band and polarization (MT) is shown in Figure 11. For XHV, CHH, and LVV, the number of observations was insufficient to create a multi-temporal combination. The combination of multiple estimates from a given frequency band is shown in Figure 12 (MTP retrieval). The agreement between in situ and retrieved stem volume for a combination of estimates from images acquired at two frequencies and three frequencies is shown in Figures 13 and 14, respectively.

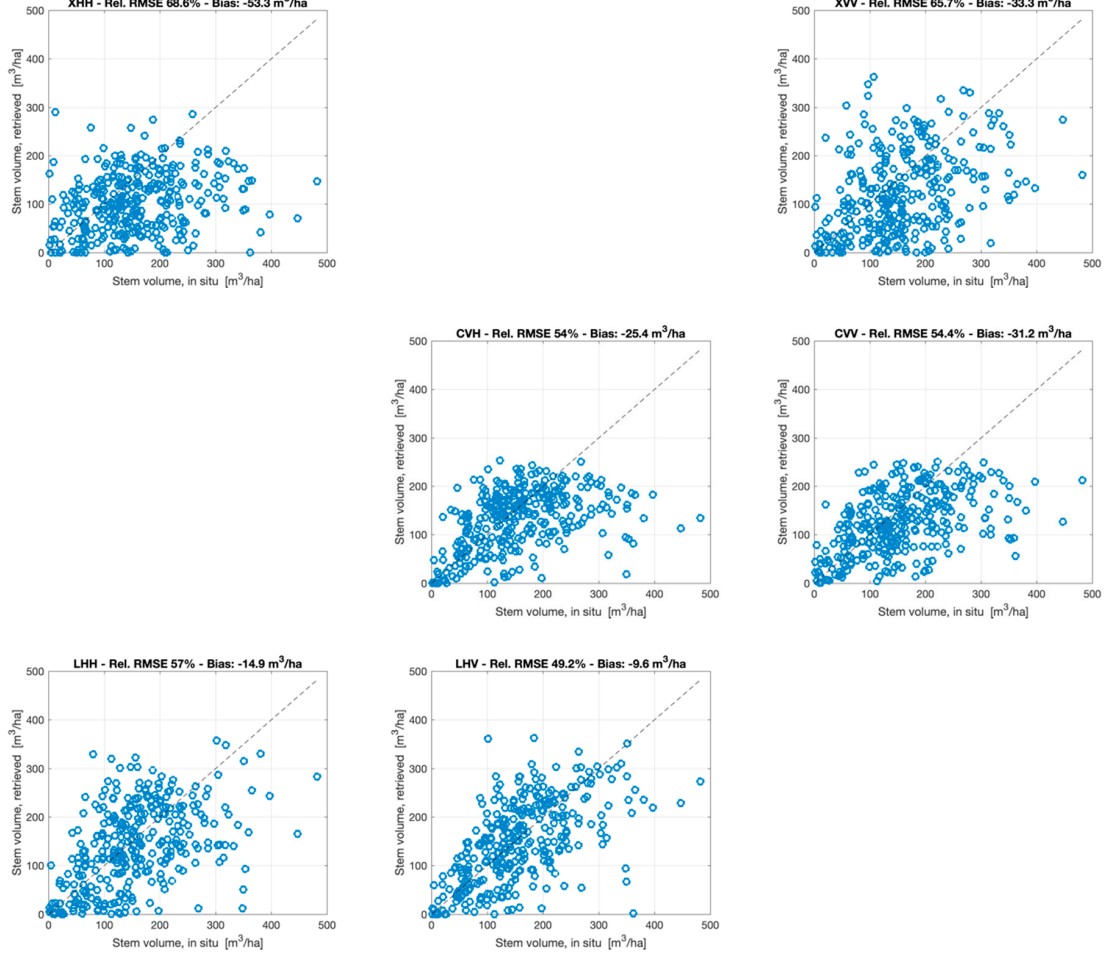

**Figure 11.** Scatter plots of measured and retrieved stem volume at Krycklan for a multi-temporal combination of estimates obtained at a given frequency band and polarization.

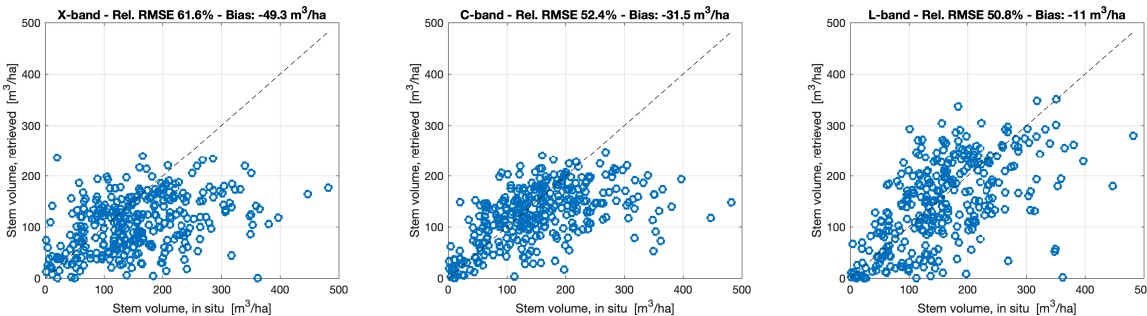

**Figure 12.** Scatter plots of measured and retrieved stem volume at Krycklan for a multi-temporal and multi-polarization combination of estimates obtained at a given frequency band.

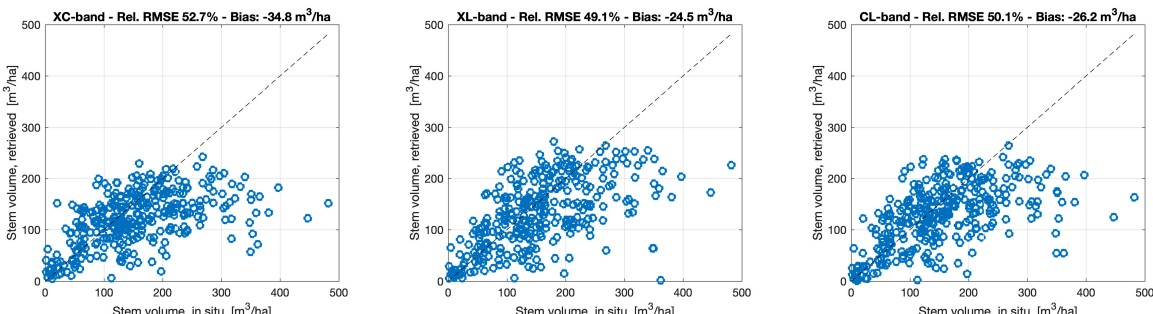

**Figure 13.** Scatter plots of measured and retrieved stem volume at Krycklan for a combination of estimates obtained from all images acquired at two frequency bands.

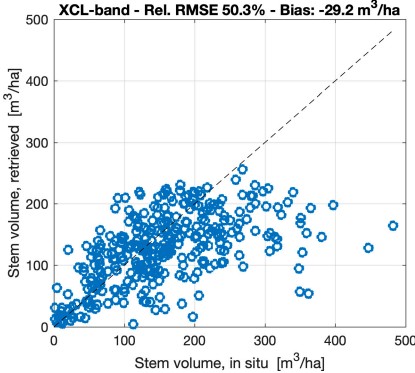

**Figure 14.** Scatter plot of measured and retrieved stem volume at Krycklan for a combination of all estimates obtained at X-, C-, and L-band.

*5.4. Assessing the Multi-Frequency Retrieval of Stem Volume*

The retrieval statistics reported in the previous Section suggested that an assessment of the multi-frequency retrieval would have been of appeal only in the case of large forest plots. The XCL-bands retrieval was, therefore, tested by sorting the 48 0.5-ha forest field inventory plots at Remningstorp for increasing stem volume and including each sample alternately in either the training set or the test set. Although this was not rigorously sound from a statistical point of view, it was a safe procedure to ensure that the same distribution of stem volumes is represented in both sets. Figure 15 shows the agreement between the in situ and the retrieved stem volumes for the test set consisting of 24 large plots.

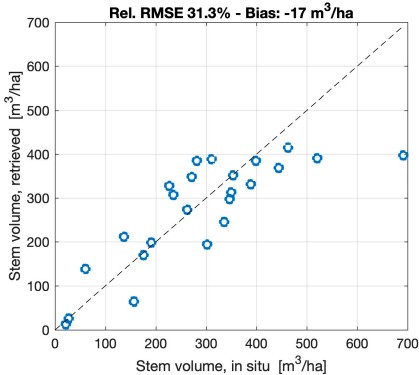

**Figure 15.** Scatter plots of measured and retrieved stem volume for 24 0.5-ha forest field inventory plots at Remningstorp. The retrieved stem volume was obtained with a combination of all estimates from SAR backscatter images acquired at X-, C-, and L-band.

Since the estimates of stem volume where obtained with a weighted average in which weights were defined on the basis of the relative RMSE (Equation (6)), plotting the individual RMSEs as a function of polarization and frequency band gave an indication of the relative importance of each image used to obtain the results in Figure 15. Figure 16 shows that the largest weights were attributed to several L-band HV-polarized images, followed by L-band co-polarized and X-band images. The Sentinel-1 C-band VV- and VH-polarized images contributed only marginally to the final stem volume estimates.

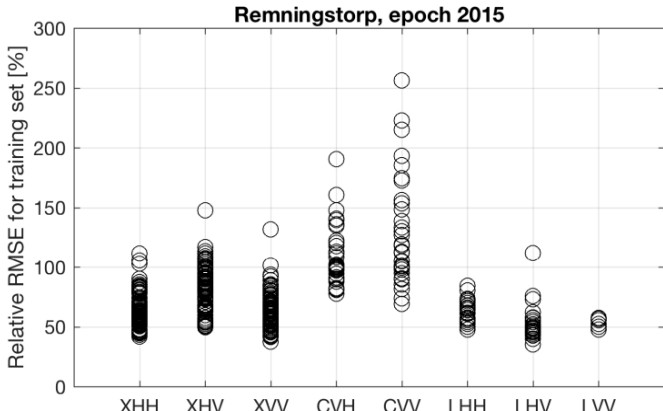

**Figure 16.** Relative RMSE or single image retrieval for the training set used to estimate stem volume shown in Figure 15. The individual relative RMSE values have been grouped in terms of frequency band and polarization.

## 6. Discussion

The availability of multi-temporal and multi-polarized SAR backscatter observations for each of the three frequency bands investigated in this study allowed for a detailed assessment of the relationship of the SAR backscatter as a function of stem volume. The relationship between SAR backscatter and stem volume from the two test sites indicated no optimal configuration in terms of frequency band, polarization, and season to estimate stem volume. At Krycklan, the sensitivity of the SAR backscatter to stem volume increased for increasing wavelength, i.e., from X- to L-band, as shown by the increasing correlation coefficient (Figure 3b) and the relationship between the two variables was always characterized by positive values (Figure 3b), i.e., a positive slope (Figure 5 and Table 5). In contrast, at Remningstorp, we observed correlation coefficients between stem volume and SAR backscatter that went from negative at X-band to positive at L-band (Figure 3a), i.e., and functional dependency with a slope that went from negative at X-band to positive at L-band (Figure 4

and Table 4). Our understanding of the observations was that, at X- and C-band, the relationship between the SAR backscatter and stem volume was affected by forest structural properties as well as additional terms that can be considered site-specific but could not be clearly identified (Figures 4 and 5). The contribution of the backscatter from the soils was stronger at Remningstorp than at Krycklan, as evidenced by the higher backscatter in low stem volume forest compared to high stem volume forest (Figure 4) where it can be reasonably assumed that the proportion of backscatter from the soil under the canopy is of minor relevance. The assumption according to which the wetter soils at Remningstorp explain the higher backscatter could neither be confirmed or rejected because of the unavailability of soil moisture measurements at the time of image acquisition. Interestingly, the largest dynamic range was observed during winter-time (Table 4) when the moisture content of the soil should be lowest because of frequent periods of frozen conditions. We may hypothesize that frozen soils allow for an increased penetration and the backscatter originates deeper in the soil. This assumption would need to be confirmed by measurements, which are currently being taken with a tower-based scatterometer and in situ observations [35]. At L-band, the relationship between SAR backscatter and stem volume seemed to be consistent among test sites and in line with previous experimental results [7]. In addition, the information provided by VV-polarization appears to be negligible when HH- and HV-polarized data are available (Figures 4 and 5). This behavior was frequency-specific, since at C- and X-band we could not observe a difference in terms of sensitivity of the SAR backscatter to stem volume at HH- and VV-polarization (Figures 4 and 5).

The RMSE values reported in Tables 4 and 5 for individual images confirm that a retrieval based on a single observation does not perform well and advocate for a combination of estimates, regardless of frequency band, polarization, and time of the year when the image was acquired. Nonetheless, there are important differences in terms of retrieval error, which can be explained in terms of the dynamic range, i.e., the sensitivity of the SAR backscatter to stem volume for a given frequency band, polarization and set of environmental conditions (Figure 6). At both sites, the most favorable configuration to retrieve stem volume corresponded to L-band and HV-polarization (Figure 6). Thereafter, X-band data proved to be more suited to retrieve stem volume at Remningstorp but not at Krycklan, where also C-band outperformed the retrieval based on X-band data (Figure 6 and Tables 4 and 5). In addition, we observed lower relative RMSEs at Remningstorp than at Krycklan (Figure 6), which could be explained by the larger size of the reference inventory plots.

As a consequence of the site-specific relationship between SAR backscatter and stem volume, the accuracy of the retrieved stem volume differed at the two sites when combining multiple estimates. At Remningstorp, the strongest agreement in the case of a multi-temporal combination was obtained at X-band for co-polarized data, at L-band for HV-polarization and at C-band for VH-polarization (Figure 7). The scatter plots of in situ and retrieved stem volumes presented light asymmetry caused by the limited sensitivity of the SAR backscatter with respect to stem volume in high biomass forest. Despite the larger number of observations available at C-band compared to X- and L-band (Table 1), the retrieval with C-band data performed poorly, particularly at VV-polarization due to the overall weak sensitivity of the backscatter to stem volume (see Figure 4 and Figures S3 and S4). The combination of estimates obtained from all images acquired at a given frequency band (MTP combination) shown in Figure 8 consolidated the results obtained with the MT combination but did not improve estimates compared to the best single-polarization case shown in Figure 7. Because of the reasonable sensitivity of X- and L-band backscatter to stem volume (Figure 4), any of the dual-frequency combinations (MTPF) performed well and slightly improved the retrieval with respect to a combination of estimates for a single frequency band (Figure 9). Combining estimates from data acquired at all three frequencies resulted in the strongest agreement between in situ and retrieved stem volumes with a 32.8% relative RMSE and a bias of $-20.4$ m$^3$/ha (Figure 10), although C-band data were marginal to the result. The worse performance of the retrieval for pine and broadleaved forests than spruce forests (Figure 10) was explained by the fact that the training set was dominated by inventory plots containing spruce forests; given the different properties of the backscatter in pine and spruce forest at Remningstorp

as discussed in Section 5, the results indicate that with a model tuned to a spruce type of forest, the retrieval did not perform equally well in other forest types, thus suggesting that models should be made adaptive to the forest structure to avoid biases in the retrieved stem volumes. This aspect has been investigated more specifically in [13] where it was proven that species-specific modeling and retrieval could improve retrieval accuracy compared to a generic model applied to all species.

Differently than at Remningstorp, the performance of the multi-temporal combination at Krycklan improved with increasing wavelength. The scatter plots did not show remarkable differences for co- and cross-polarized data, which is probably due to the size of the inventory plots causing a large spread of the backscatter observations when related to stem volume. The combination of all data acquired at one frequency (MTP) showed slight improvement with respect to the single polarization retrievals (Figure 12). The agreement between in situ and retrieved stem volume increased for increasing wavelength. The retrieval further improved when combining estimates of stem volume from data acquired at two or three frequencies (MTPF) (Figures 13 and 14). The relative RMSE for the retrieval based on X-, C- and L-band was almost twice the value obtained at Remningstorp (50.3% vs. 32.8%). The difference shall be seen as a consequence of the much smaller size of the forest field inventory plots (0.03 ha vs. 0.5 ha) and confirms that the retrieval of biomass with SAR backscatter performs poorly when assessed at the level of inventory units having a size comparable to the pixel of the SAR image.

The assessment of the retrieval accuracy with a multi-frequency retrieval based on the X-, C-, and L-band SAR backscatter datasets at Remningstorp revealed strong agreement of in situ and retrieved stem volume throughout the range of values represented (Figure 15). The relative RMSE of 31.3% and the bias of $-17$ m$^3$/ha were strongly influenced by the plot with the highest stem volume for which the retrieval predicted slightly less than 400 m$^3$/ha. For this plot, the backscatter was consistently in the range of values observed for forests with a stem volume of 300–400 m$^3$/ha. The inventory measurements, however, did not reveal any similarity in terms of basal area, tree density and tree height so that the reason for such result remains unexplained. When neglecting this plot, the relative RMSE was 25% with a negligible bias. The multi-frequency retrieval results obtained here outperform single frequency retrievals undertaken at X-, C-, and L-band in this study (Figure 16) as well as in previous studies when considering retrieval accuracies for similar types of reference data (Table 6). Only at P-band and at the VHF frequency band did the retrieval perform with higher accuracy (Table 6), as also demonstrated in [13].

**Table 6.** Overview of previous studies dealing with retrieval of forest stem volume and above-ground biomass at the two test sites using SAR backscatter. Studies based on interferometric, polarimetric and polarimetric interferometric SAR observables are not considered because off-topic.

| Band | Sensor | Test Sites | Reference Data | Remarks | Reference |
|------|--------|-----------|----------------|---------|-----------|
| L | ALOS PALSAR | Remningstorp and Krycklan | Hectare-scale stands | Multi-temporal dataset RMSE: 35% and 44%. | [7] |
| L | E-SAR | Remningstorp | Sub hectare-scale stands Laser scanning data and 10- radius inventory plots | Single-image retrieval RMSE: 31–46% | [18] |
| P | E-SAR | Remningstorp | Sub hectare-scale stands Laser scanning data and 10- radius inventory plots | Single-image retrieval RMSE: 18–27% | [18] |
| P | E-SAR | Remningstorp and Krycklan | Hectare-scale stands | RMSE: 28–42% at Krycklan. RMSE of 22–33% using the backscatter model developed at Krycklan | [19] |
| VHF | CARABAS | Remningstorp | Hectare-scale stands | Multiple viewing directions RMSE: 11–25% | [36] |

## 7. Conclusions

This study aimed at investigating the retrieval of forest stem volume in Swedish forests using short wavelength SAR backscatter data. The relevance of this study lies in the current scenario of observations by spaceborne sensors at X-, C-, and L-band. Thanks to the availability of multiple observations at each frequency, it has been demonstrated that the retrieval of forest stem volume profits from the availability of multiple observations acquired at the three frequencies. The retrieval improved with respect to single image and multi-temporal single frequency retrievals because of the larger array of observations available, which were optimally combined to maximize the information content on stem volume across all frequencies. The weighted average of estimates obtained from each of the images forming the multi-frequency dataset was key to achieving an improved estimate with respect to the input data. Differently than stacking observations at a single frequency, where specific systematic errors and uncertainties embedded in the signal still affect the retrieved biomass, the combination of estimates of biomass obtained at different frequencies allows for compensating frequency-specific nuances.

The retrieval was shown to perform better at the stand (i.e., hectare) level; a retrieval error of approximately 25% was achieved with X-, C-, and L-band. Retrieval of biomass at plot level performed poorer because at such scale, the link between the biomass measured in a plot and how this biomass impacts the signal backscattered to the radar is weak. When using a multi-frequency retrieval based on the inversion of the WCM, we achieved a retrieval accuracy of 40% at best. By selecting different models for the inversion depending on SAR frequency, an error of about 30% was achieved with C-, L-, and P-band.

Our results do not seem to support the common understanding that the retrieval of forest biomass improves with increasing wavelength. The different environmental settings, forest structural characteristics and forest management of the two sites significantly shaped the relative contribution of different scattering mechanisms as function of increasing biomass, not allowing for a definite conclusion on which frequencies should be preferred and what would be the minimum number of observations required to achieve a certain accuracy. However, having available multi-temporal observations at multiple frequencies, the combination of many observations in time and in frequency allows for the most accurate retrieval achievable with short wavelength SAR backscatter data. This result appears to be independent from the specific structural properties of the forest. Nevertheless, even if in combination, short wavelength SAR backscatter observations are not able to achieve the highest possible retrieval accuracy, as demonstrated for the same sites in the case of long wavelength SAR systems operating at P- or VHF-band.

Looking at current and forthcoming spaceborne SAR missions, multi-frequency observations are a reality and their number will further increase in the next decade with the launch of missions carrying onboard radar instruments operating between X- and P-band. Given the encouraging results obtained in this study on the complementarity of short-wavelength multi-frequency SAR data to retrieve forest biomass, we also see a need to further investigate retrieval approaches in other forest biomes to improve the understanding of multi-frequency SAR data towards the retrieval of forest biomass.

**Supplementary Materials:** The following are available online at http://www.mdpi.com/2072-4292/11/13/1563/s1. Figure S1: Estimates of $\sigma^0_{gr}$ and $\sigma^0_{veg}$ for each X-band co-polarized image over Remningstorp (left: TerraSAR-X HH-polarization; right: TerraSAR-X VV-polarization) together with profiles of daily temperature extremes, precipitation and snow depth. The panel at the bottom illustrates the retrieval error for the model tested with the same samples used for the training. Figure S2: Estimates of $\sigma^0_{gr}$ and $\sigma^0_{veg}$ for each X-band HV-polarized image over Remningstorp (TerraSAR-X) together with profiles of daily temperature extremes, precipitation and snow depth. The panel at the bottom illustrates the retrieval error for the model tested with the same samples used for the training. Figure S3: Estimates of $\sigma^0_{gr}$ and $\sigma^0_{veg}$ for each C-band VV-polarized image over Remningstorp together with profiles of daily temperature extremes, precipitation and snow depth. The panel at the bottom illustrates the retrieval error for the model tested with the same samples used for the training. Figure S4: Estimates of $\sigma^0_{gr}$ and $\sigma^0_{veg}$ for each C-band cross-pol image over Remningstorp together with profiles of daily temperature extremes, precipitation and snow depth. The panel at the bottom illustrates the retrieval error for the model tested with the same samples used for the training. Figure S5: Estimates of $\sigma^0_{gr}$ and $\sigma^0_{veg}$ for each L-band HH-polarized

image over Remningstorp together with profiles of daily temperature extremes, precipitation and snow depth. The panel at the bottom illustrates the retrieval error for the model tested with the same samples used for the training. Figure S6: Estimates of $\sigma^0_{gr}$ and $\sigma^0_{veg}$ for each L-band VV-polarized image over Remningstorp together with profiles of daily temperature extremes, precipitation and snow depth. The panel at the bottom illustrates the retrieval error for the model tested with the same samples used for the training. Figure S7: Estimates of $\sigma^0_{gr}$ and $\sigma^0_{veg}$ for each L-band HV-polarized image over Remningstorp together with profiles of daily temperature extremes, precipitation and snow depth. The panel at the bottom illustrates the retrieval error for the model tested with the same samples used for the training. Figure S8: Estimates of $\sigma^0_{gr}$ and $\sigma^0_{veg}$ for each X-band co-polarized image over Krycklan (left: TerraSAR-X HH-polarization; right: TerraSAR-X VV-polarization) together with profiles of daily temperature extremes, precipitation and snow depth. The panel at the bottom illustrates the retrieval error for the model tested with the same samples used for the training. Figure S9: Estimates of $\sigma^0_{gr}$ and $\sigma^0_{veg}$ for each C-band VV-polarized image over Krycklan together with profiles of daily temperature extremes, precipitation and snow depth. The panel at the bottom illustrates the retrieval error for the model tested with the same samples used for the training. Voids in the temporal profiles represent image acquisition dates with only partial coverage of the test site. Figure S10: Estimates of $\sigma^0_{gr}$ and $\sigma^0_{veg}$ for each C-band VH-polarized image over Krycklan together with profiles of daily temperature extremes, precipitation and snow depth. The panel at the bottom illustrates the retrieval error for the model tested with the same samples used for the training. Voids in the temporal profiles represent image acquisition dates with only partial coverage of the test site. Figure S11: Estimates of $\sigma^0_{gr}$ and $\sigma^0_{veg}$ for each L-band HH-polarized image over Krycklan together with profiles of daily temperature extremes, precipitation and snow depth. The panel at the bottom illustrates the retrieval error for the model tested with the same samples used for the training. Voids in the temporal profiles represent image acquisition dates with only partial coverage of the test site. Figure S12: Estimates of $\sigma^0_{gr}$ and $\sigma^0_{veg}$ for each L-band HV-polarized image over Krycklan (left: ALOS-1; right: ALOS-2) together with profiles of daily temperature extremes, precipitation and snow depth. The panel at the bottom illustrates the retrieval error for the model tested with the same samples used for the training. Voids in the temporal profiles represent image acquisition dates with only partial coverage of the test site.

**Author Contributions:** Conceptualization, M.S., O.C. and U.W.; methodology, M.S. and O.C.; validation, M.S. and J.E.S.F. writing—original draft preparation, M.S.; writing—review and editing, M.S., O.C., J.E.S.F. and U.W.; supervision, M.S.; project administration, U.W.

**Funding:** This research was funded by the European Space Agency (Contract 4000115192/15/NL/AF/eg).

**Acknowledgments:** We are thankful to ESA's Technical Officer, Björn Rommen, for the support and the advices during the project activities. The comments by the three anonymous Reviewers improved the clarity of the text significantly. TerraSAR-X data were obtained through a Science proposal lead by Chalmers University of Technology (Proposal XTI_VEGE6857, 3D Forest Structure and Biomass Retrieval in Boreal Forests). We are thankful to Maciej Soja for making the data available. Sentinel-1 data were obtained from the European Space Agency through the Sentinel- Science Hub. ALOS-2 PALSAR-2 data were obtained from the Japan Aerospace Exploration Agency (JAXA) through a science proposal in the framework of Research Announcement 4 (RA-4). Weather data were collected from archives of the Weather Underground website (http://www.wunderground.com).

**Conflicts of Interest:** The authors declare no conflict of interest. The funders had no role in the design of the study; in the collection, analyses, or interpretation of data; in the writing of the manuscript, or in the decision to publish the results.

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
