# Peer review of "Complementarity of X-, C-, and L-band SAR Backscatter Observations to Retrieve Forest Stem Volume in Boreal Forest"

_remotesensing, doi:10.3390/rs11131563_

Round 1
Reviewer 1 Report
Review of the manuscript Maurizio Santoro et al., “Complementarity of X-, C- and L-band SAR backscatter observations to retrieve forest stem volume in boreal forest”.
The authors investigated the retrieval of forest stem volume in Swedish forests using SAR backscatter data acquired by satellite borne sensors at X-, C- and L-band. Before retrieval, the relationship between SAR backscatter and forest stem volume in X-, C- and L-band observations were analyzed. Retrieval of stem volume was implemented by using weighted average of estimates, water cloud model (WCM) and multiple observations (i.e., multi-temporal (MT), multi-polarized and multi-temporal (MTP), Multiple frequency bands, multi-polarization and multi-temporal SAR data). To assess the performance of the proposed method, the retrieval results are compared against the in situ measurements. It achieved an accuracy that was superior to values obtained at a single frequency. The experimental results are valuable and the work has some interesting points. Before considering the publication, I have some minor issues and major concerns on the manuscript and results that the authors should address carefully.
Minor Issues:
i. In Abstract Section, line 29, page1: I didn’t see the value “32.0%” and “46.8%” in the part of result, discussion or conclusion, also cannot get these information from figures.
ii. In Abstract Section, line 33-34, page1: Choose keywords correctly, reduce the number of the keywords and keep the most importance.
iii. Line 68-92, Page 2: I suggest the author should add the largest research literature, introduce detail the deficiencies by multi-frequency observations to retrieve biomass in recent and then talk about the main innovation and contribution.
iv. In the section 3, page 3: I suggest the author add the incidence angle information of X,C and L band in different modes in table 1 and 2.
v. Recommend to increase the size of figures in this manuscript (such as Figure 3, Figure 4, etc.) as well as the font size in the figure (such as the Figure 2).
vi. In Section 5.2, line 399, page11: the word “supplement” should not be capitalized.
vii. In Section 5.3, line 454-457, page15: The format, such as the font size and the margin, of this paragraph should be same as the main body of the manuscript.
viii. In Section 6, line 580, page19: the word “in situ” should be italic.
ix. Line 399, 402, 406, 408,411-413, 46, 442, and etc. The authors mentioned “Figure S#“, where I do not find these figures in this manuscript.
x. There are problems with the typesetting of the equations. Takes the equation 5 and 6 as an example. what is the meaning of point ‘.’ in all the equations?
Major Concerns:
i. The authors could consider adjusting the structure of the Section 3 and Section 5. For example, in Section 3, the authors could add subheadings in this part for the authors devoted extensive coverage to introduce the SAR data processing; In Section 5.3, the authors has introduced the retrieval of stem volume using multiple observations, where in Section 5.4 “Multi-frequency retrieval of stem volume” is introduced again, which is duplicate and redundant.
ii. Section 5.1, line 317-318: The authors introduced “decreasing SAR backscatter with increasing stem volume, with values between -0.5 and -0.1”. Where in the Figure 3(a), CVH and CVV the correlation coefficient is between -0.5 and 0. I can not find any high correlation between C-band backscatter and stem volume.
iii. Section 5.1, line 322-323, 335-337: the author mentioned that “Somewhat stronger sensitivity of the SAR backscatter to stem volume …” and “…with slightly stronger sensitivity of the…”, I think we could not judge which one has stronger sensibility of SAR backscatter to stem volume just from figures, when comparing frozen conditions and unfrozen conditions (see dotted rectangle area). Here, the authors can calculate the average value of correlation coefficient in frozen conditions and unfrozen conditions, respectively.
Figure 3. Correlation coefficient of SAR backscatter and stem volume as a function of day-of-year, (DOY) at Remningstorp (a) and Krycklan (b). Colors refer to the minimum temperature on the day of image acquisition. Crosses are blue or red depending whether the temperature was below or above 3°C.
iv. Line 339-343, part of “5.2 Estimates of WCM parameters”: I suggest the authors to provide the number of training points. Besides, the authors mentioned that “For C- and L-band, values published in previous studies for the two sites were used, i.e., 0.0055 and 0.0042 ha/m3, respectively [7,27]”. I wondered that whether the empirical parameter is same for different polarization (e.g., HH, VV and HV), and why the author selected only one empirical value for different polarizations. The sensitivity is also different in different polarization of L-band and C-band.
v. Line 354-360, part of “5.2 Estimates of WCM parameters”: The relationship between SAR backscatter and stem volume changed from decreasing to increasing from X- to L-band in Remningstorp site, where the backscatter always increased with increasing stem volume in Krycklan site. The authors interpret that this difference is own to the soil properties, such as surface roughness and soil moisture. As we all know, the X-band and C-band microwave both have limited penetrating ability, does the behavior of the changing of SAR backscatter have relationship with vegetation water content (VWC)? If not, why the authors ignore the influence of the VWC?
vi. In section 5.3, the author shows the scatter plots of in situ and retrieved stem volume at remningstorp in the case of MT, MTP and MTPF. I suggest the author fails to detailly explain the accuracy is poor in C band, and include C band at two frequency bands. And why the VV backscattering coefficient is obvious or sensitive than HV polarization, however, the RMSE is poor, the opposite result is observed in X and L polarization.

Author Response
Dear Reviewer
Thanks you for the set of interesting remarks. We have dealt with each and provided a point-by-point reply in the attached document.
Sincerely
Maurizio Santoro

Reviewer 2 Report
In this paper, authors explored the feasibility of collaborative estimation method of forest stem volume using X-band, C-band and L-band SAR data. The manuscript proposed a method for weighted average of water cloud model estimation results under different observation conditions. Based on two test sites, the estimation performances of SAR data with different wavelength and wavelength combinations was analyzed. The topic is timely and relevant and the experimental design is sound.
However, I have two suggestions for this paper.
1) The authors compared the estimation results on different wavelength and wavelength combinations in both test site. From the results in the second test site(Krycklan),it can be seen that the accuracy combining all estimates obtained at X-, C- and L-band is lower than the best accuracy of the estimation results obtained at a given wavelength and polarization. I am wondering that, if they can explain the advantages of the weighted average method of estimates from a statistical point of view , and under what circumstances this method can obtain better accuracy than the best estimates using a single wavelength and single polarization SAR data.
2) The two accuracy indicators adopted in this paper are relative RMSE and bias, both of them are indexes for estimation error of the method. The fitting performance of the method was not fully evaluated. It is better if the authors can add coefficient of determination (R2) as accuracy indicator, which is convenient for readers to have a quantitative judgment on the fitting performance of the method.
Author Response

(The authors gave the same response as above.)

Reviewer 3 Report
I have read the manuscript entitled "Complementarity of X-, C- and L-band SAR 3 backscatter observations to retrieve forest stem 4 volume in boreal forest" with great interest. It seems interesting research to select the sensor and polarization of the SAR data to estimate biomass/stem volume.
I have the following specific questions:
1) Authors study is site specific so how this study can contribute significantly in the selection of the SAR sensor to estimate stem volume
2) The author didn't mention about limitations of this study so he should specify.
3) The author didn't mention details about the speckle filtering. The author should specify the filtering techniques and whether they had used the same filter for all the data or different filter for different data
4) The author should give more details about the importance of ENL.
5) The author mentioned that they selected hyptothetically homogeneous scattering. Please elaborate
6) The author has mentioned that backscattering varies with temperature and other meteorological parameters but X-band doesn't show the change in the correlation coefficient fig 3a.
7) why the author just discussed about temperature effects not rainfall and snow density effects on backscattering?
8) Usually, HV/VH play a significant role in biomass estimation but its not clear using sentinel data
9) why there is no negative relationship of XHH, XHV, XVV polarization in figure 3b
10) The author mentioned that "scatter plots did not show remarkable differences for co- and cross-polarized data". Please elaborate more because previous studies clearly show a strong relationship with cross-pol as compared to co-pol.
11) Did the author notice any saturation if use low-frequency X-band as compared to L-band in the boreal forest?
Author Response

(The authors gave the same response as above.)

Reviewer 4 Report
Complementarity of X-, C- and L-band SAR backscatter observations to retrieve forest stem volume in boreal forest
The work seeks in characterizing the relationship between multiple frequencies SAR backscattering information and stem volume of the forests at two sites in Sweden. The relations showed site dependent relation for X and C band while the L-band showed conventional characteristics for both sites. Combination of all three frequencies improved the retrieval accuracy compared to other combinations.
It is interesting to see the characteristics of different frequency SAR to the relation with forest attributes. While more studies are also seeking for the integration of multi frequency SAR images, this work shows the up-to-date information for when using all XCL bands. I am although curious for few things. One is, what was the trend of the backscattering responses, as a function of the local slope angle. From the processing of the SAR flowchart, it seems that all aspect is considered for the pre-processing, but just wanted to make sure that there is no trend for this (especially for trends when dividing into forward slopes (slope facing sensor direction) and backward slopes (slopes facing away from sensor direction)). Another is, about the decreasing trend of the backscatter, where it is mentioned that forest structural properties was affecting this. Can this be understood as signal attenuation is occurring during the observation on higher volume areas? I wonder if there could be some discussion little bit more about this relating with other similar works backscattering trend, if there is any. Last thing is in the future, it would be interesting if we could see something like random forest regression using all three frequencies for retrieval of the forest attributes. Thank you.
Small comment
- Table 4 shows like XHH, XHV, XVV… Table 5 shows HH, VV… show it either way.
Author Response

(The authors gave the same response as above.)
